# Influenza A vs. COVID-19: A Retrospective Comparison of Hospitalized Patients in a Post-Pandemic Setting

**DOI:** 10.3390/microorganisms13081836

**Published:** 2025-08-06

**Authors:** Mihai Aronel Rus, Daniel Corneliu Leucuța, Violeta Tincuța Briciu, Monica Iuliana Muntean, Vladimir Petru Filip, Raul Florentin Ungureanu, Ștefan Troancă, Denisa Avârvarei, Mihaela Sorina Lupșe

**Affiliations:** 1Department of Infectious Diseases, Iuliu Hatieganu University of Medicine and Pharmacy, Iuliu Moldovan Street, No. 23, 400348 Cluj-Napoca, Romania; aronelrus@gmail.com (M.A.R.);; 2Department of Medical Informatics and Biostatistics, Iuliu Hatieganu University of Medicine and Pharmacy, Pasteur Street, No. 6, 400349 Cluj-Napoca, Romania; 3Clinical Hospital for Infectious Diseases, Iuliu Moldovan Street, No. 23, 400348 Cluj-Napoca, Romania

**Keywords:** influenza, COVID-19, inflammation, prognosis, severe infection

## Abstract

In this paper we aimed to compare seasonality, clinical characteristics, and outcomes of Influenza A and COVID-19 in the context of influenza reemergence and ongoing Omicron circulation. We performed a retrospective comparative analysis at the Teaching Hospital of Infectious Diseases in Cluj-Napoca, Romania. We included adult patients hospitalized with Influenza A or COVID-19 between 1 November 2022 and 31 March 2024. Data were collected on demographics, clinical presentation, complications, and in-hospital mortality. We included 899 COVID-19 and 423 Influenza A patients. The median age was 74 years for COVID-19 and 65 for Influenza A (*p* < 0.001). The age-adjusted Charlson comorbidity index was higher in COVID-19 patients (5 vs. 3, *p* < 0.001). Despite this age gap, acute respiratory failure was more common in Influenza A (62.8% vs. 55.7%, *p* = 0.014), but ventilation rates did not differ significantly. Multivariate models showed Influenza A was associated with increased risk of intensive-care unit (ICU) admission or ventilation, whereas older COVID-19 patients had higher in-hospital mortality (5.67% vs. 3.3%, *p* = 0.064). Omicron COVID-19 disproportionately affected older patients with comorbidities, contributing to higher in-hospital mortality. However, Influenza A remained a significant driver of respiratory failure and ICU admission, underscoring the importance of preventive measures in high-risk groups.

## 1. Introduction

Influenza and COVID-19 are two viral infections with significant morbidity and mortality all over the world. In Europe, seasonal outbreaks of influenza happen during the cold season [1]. During the 2020–2021 season, when COVID-19 restrictions were effective, the incidence of influenza was peculiarly low, but starting from the 2022–2023 cold season, the number of flu cases resurged to pre-pandemic patterns, with more than 200,000 confirmed cases a year in Europe, a majority being caused by type A influenza virus [2,3]. COVID-19, caused by the novel coronavirus SARS-CoV-2, caused more than 270 million cases in Europe [4] since January 2020, with more than 2.2 million deaths [5], higher than estimated deaths from influenza in the past 20 years [6]. The Omicron variant of SARS-CoV-2, dominant worldwide since 2022, showed increased transmissibility, displayed a decrease in severity when compared to previous variants [7], and caused consecutive waves across the globe [8]. Both conditions manifest as respiratory tract infections, sharing significant similarities in clinical presentation and transmission modalities, with seasonal peaks potentially overlapping during the cold season [9]. Omicron COVID-19 and influenza remain the most frequent viral respiratory tract infections in Europe, especially during the cold season [10]. Risk factors for severe disease are well documented and largely similar for both infections [11,12]. However, most published studies focused on SARS-CoV-2 variants that preceded Omicron, and updated comparative information data on hospitalized patients with these two major viral illnesses remain limited.

The aim of this paper was to compare the two most prevalent viral respiratory tract infections (type A influenza and Omicron COVID-19), in terms of clinical characteristics, risk factors for severe disease, and unfavorable outcomes, such as intensive care unit (ICU) admission, need for ventilation, apparition of complications, and in-hospital mortality in a post-pandemic context, using data from patients hospitalized in a Romanian infectious diseases hospital between November 2022 and March 2024.

## 2. Materials and Methods

We performed a retrospective comparative analysis with data from patients hospitalized in the Teaching Hospital of Infectious Diseases from Cluj-Napoca, Romania. The data were retrieved from the electronic database of the hospital. We included patients ≥18 years old hospitalized for COVID-19 or Influenza A from 1 November 2022 until 31 March 2024. We confirmed influenza or COVID-19 with rapid antigenic testing or polymerase chain reaction (PCR) testing, according to WHO case definitions.

Subjects were initially screened with ICD-10 coding, using J10.0, J10.1, J10.8, J11.0, J11.2, and J11.8 for influenza and B97.2 and U07.1 for COVID-19, respectively. Tests used for diagnosis were Vitrotrack Influenza A + B rapid antigenic test (87.2% sensitivity and 94.5% specificity for Influenza A), Standard Q COVID-19 antigenic test (97.12% sensitivity, 100% specificity), GeneXpert-Cepheid@ real-time PCR testing (98.9% sensitivity for Influenza A; 98.4% sensitivity for influenza B; 97.5% specificity for Influenza A; 99.3% specificity for influenza B; 97% sensitivity and specificity for SARS-CoV-2), DiagCORE-Qiagen@ RT-PCR Influenza A and B (97.84% sensitivity and 99.45% specificity) Seegene AllplexTM RT-PCR SARS-CoV-2 assay (100% sensitivity, 98% specificity), NeuMoDxTM RT-PCR SARS-CoV-2 assay (100% sensitivity, 98% specificity).

Patients positive for more than one viral etiology, asymptomatic patients, patients with B influenza, and patients <18 years old were excluded. Based on country-level epidemiological data, all COVID-19 infections during the study period were attributable to Omicron variant of SARS-CoV-2, and no significant shifts in clinical presentation or management occurred. Similarly, Influenza A cases were caused by antigenically similar strains. Therefore, we treated the timeframe as a stable clinical context and did not include time as a covariate in multivariate models. The case definitions were according to WHO recommendations.

For our analysis, we made a database, and for each subject included in the analysis, we extracted variables such as age, gender, length of hospitalization, symptoms, intensive care unit (ICU) admission, comorbidities (age-adjusted Charlson comorbidity index (ACCI) and Charlson comorbidity index (CCI) were calculated), laboratory and radiology parameters, complications, and in-hospital mortality. Information related to seasonal influenza and COVID-19 vaccination was also retrieved.

The study was conducted in accordance with the Declaration of Helsinki and was approved by the Ethics Committee of the Teaching Hospital of Infectious Diseases Cluj-Napoca (5824/3 April 2024).

Data collection and management was performed with Microsoft Excel and statistical analysis with R. Categorical data were presented as counts and percentages. Nonnormally distributed quantitative data were presented as medians and the first and the third quartiles. Comparisons between two groups concerning categorical variables were performed with the Chi-squared test or Fisher exact test. Comparisons between two groups concerning nonnormally distributed quantitative variables were performed with the Wilcoxon rank-sum test. To compare COVID-19 with influenza more in depth, multivariate models were fit. To prevent overfitting, we calculated the maximum number of variables that would be appropriate to include in the models by dividing the number of participants in the less represented category of the dependent variable to ten. For death outcome, we found a maximum of six variables. The model contained as variable of interest the disease (COVID-19 compared to influenza) and a list of pre-defined predictors based on clinical judgment: age, sex, ACCI, C-reactive protein (CRP), and creatinine. Due to finding a multicollinearity (by assessing variance inflation factor and the correlation matrix) between age and ACCI, we excluded the age variable from the model, being reassured by the fact that ACCI includes age in its calculation. For the models predicting intensive care unit admission, continuous positive airway pressure ventilation (CPAP), and intubation, a larger number of predictors was allowed. For all models, we tested the linearity of the continuous predictors with the logit of the model, using a generalized additive model with a logit link function, and we found that all continuous predictors had nonlinear relationships. Thus, we presented two models for each dependent variable: one generalized additive model with splines for continuous variables, to correctly capture the functional form (and control for) of the confounding variables, and one multivariate logistic regression model with the continuous variables dichotomized by their medians, to explore clinically their association with the dependent variable. The rationale for using both models was to balance statistical rigor and clinical interpretability: spline-based models accurately capture nonlinear effects, while dichotomized models allow clinicians to understand odds ratios based on common clinical thresholds. This dual approach enhances both analytic accuracy and translational relevance. For all logistic regression models, the goodness of fit was tested with Hosmer and Lemeshow test.

## 3. Results

### 3.1. Seasonality and Cohort Characteristics

According to inclusion criteria, we identified 1322 hospitalized patients: 899 COVID-19 patients and 423 type A influenza patients. During the evaluated timespan, there were 26 influenza B cases and 9 cases of coinfection with SARS-CoV-2, which were excluded from the analysis.

Almost all cases of flu (*n* = 418, 98.8%) were hospitalized in a November–March (Week 46–Week 12) interval, while outbreaks peaked in January and ended in March; very few cases (*n* = 5, 1.2%) were admitted during warmer months. The number of COVID-19 admissions was more than double; although the latter infection was also prevalent during the cold season, hospitalizations occurred all year round and a significant proportion of COVID-19 cases (*n* = 382, 42.49%) happened in the April–October timespan, with very few cases admitted during summer. Flu outbreaks were shorter than COVID-19 waves and their peaks did not overlap. A representation of weekly hospitalizations is shown in Figure 1.

The median age of Influenza A patients was 65 years old (range 18–100) and the median age for COVID-19 patients was 74 years old (range 18–103), *p* < 0.001. The gender distribution was 255 female patients (60.28%) with Influenza A and 501 (55.72%) female patients with COVID-19, respectively. The ACCI for Influenza A was 3, and for COVID-19 was 5 (*p* < 0.001), but Charlson comorbidity index (does not include age) was identical between the groups. Influenza vaccination was found in 17% of patients with known status, whereas COVID-19 vaccination in 76% of patients with known status, respectively, but only 7 subjects had received the updated Omicron vaccine with more than three doses. Detailed characteristics and comorbidities of hospitalized patients are represented in Table 1.

The results of laboratory tests and radiology findings are presented in Table 2. Several differences between the groups are revealed, mainly a lower number of lymphocytes (*p* < 0.049), lower number of monocytes (*p* < 0.008), and higher LDH for Influenza A. Inflammation parameters like CRP and complete blood count (CBC)-derived NLR (Neutrophils-to-lymphocytes ratio) and SIRI (systemic inflammation response index) did not achieve statistical significance but were high in both groups. Abnormal imaging, including ground-glass opacities, viral interstitial pneumonia, or bilateral consolidation were more frequent in COVID-19, except for unilateral consolidation, which was more frequent for Influenza A.

### 3.2. Complications and Unfavorable Outcomes (ICU Admission, Mechanical Ventilation, and Death)

Severe Influenza A was diagnosed in 297 patients (70.21%); severe COVID-19 was diagnosed in 261 cases (29.03%) and critical COVID-19 in 118 patients (13.13%). Therefore, patients with criteria for severe disease were more frequent for Influenza A than for severe/critical Omicron COVID-19 (Table 3). The median age of patients with severe Influenza A was 71 years old, while Omicron patients with severe/critical disease had a median age of 78 years old (*p* < 0.010). Female sex was more prevalent for severe Influenza A (55.22%), but severe/critical Omicron patients were more often male (55.56% and 50.85% respectively, *p* = 0.016).

The two groups were identical regarding the duration of illness until hospital presentation (3 days) and median length of stay (6 days). Approximately 10% of patients needed ICU admittance, more often in the Omicron group, without statistical significance (11.79% Omicron vs. 9.46% Influenza A, *p* = 0.206); duration of ICU admission was identical for both infections, i.e., 7 days.

Complications and outcomes are presented in Table 4. Acute respiratory failure with need of oxygen supplementation was the most prevalent complication for both conditions, being more frequent for Influenza A (*p* = 0.014), but ventilatory support (both CPAP and mechanical ventilation) did not differ significantly between the groups. Acute renal failure was also more frequent in Influenza A (21.99% vs. 16.69%, *p* = 0.02). A new onset of atrial fibrillation was the most frequently diagnosed cardiovascular complication, being more often in COVID-19 patients. Embolic complications such as pulmonary embolism was encountered in 3% of COVID-19 hospitalizations. Other complications in the analysis, including myocardial infarction, were very rare in both cohorts. Antibiotic prescription was identified in a high number of patients for both conditions. In the database, there were 51 (5.67%) fatal events for COVID-19 and 14 (3.31%) for Influenza A.

The patients who died from COVID-19 were older (median 80 years-old vs. 75 years-old, *p* = 0.18), had a longer length of stay (13 days vs. 10 days, *p* = 0.823), and had a longer ICU stay (11 days vs. 8 days, *p* = 0.778) than influenza patients, but without statistical significance; the median ACCI for both groups was 7 (Table 5). There were no deaths recorded in pregnant women or solid organ transplant recipients. Laboratory parameters revealed very low lymphocyte count and very high values of inflammation markers, including NLR and C-reactive protein (Appendix A).

Multivariate regression models that compare COVID-19 and influenza in predicting intensive care unit admission, need for CPAP, or intubation were envisaged, adjusting for the following confounders: sex, ACCI, CRP, creatinine, LDH, and NLR. Upon inspection, nonlinear relationships were found between the continuous confounders. Therefore, two models were fit: one using splines for continuous predictors and another with continuous predictors dichotomized by their median values. Interestingly, when controlling confounders, the observed odds of intensive care unit admission, need for CPAP, or need for intubation were higher for infection with Influenza A virus, compared to SARS-CoV-2 virus infection, without statistical significance. In the model with splines, influenza had significantly higher odds of CPAP compared to COVID-19. Higher values of CRP, LDH, and NLR were associated with significantly higher odds of ICU admission, need for CPAP, or need for intubation, respectively. In predicting death outcome, a multivariate analyze was used, adjusted for the following confounders: sex, ACCI, CRP, and creatinine. As previously, two models were fit: one using splines for continuous predictors and another with continuous predictors dichotomized by their median values. There was no significant difference between COVID-19 and influenza in predicting death, in both models. Nevertheless, the observed odds of death in the study were higher for COVID-19 compared to influenza. Higher values of Charlson comorbidity index, CRP, and creatinine were associated with significant higher odds of death outcome (Table 6).

The result of the Hosmer–Lemeshow goodness-of-fit test, which indicates whether there are differences between the observed data and the model’s prediction, was as follows: ICU admission, *p* = 0.076; CPAP, *p* = 0.848; and intubation, *p* = 0.002. Regarding prediction, the overall percentage of correct classification was as follows: ICU admission, 83.82%; CPAP, 79.53%; and intubation, 91.92%. The area under the curve was as follows: ICU admission, 82.46 (78.51–86.4); CPAP, 79.75 (76.01–83.49); and intubation, 83.16 (78.12–88.19).

## 4. Discussion

In this paper, we conducted a comparative analysis between type A influenza, the most common type of flu in Europe [13], and Omicron COVID-19. In this database, during the analyzed timespan that included two complete influenza seasons, the number of hospitalizations caused by SARS-CoV-2 virus was approximately double. Put in perspective, Romania reported in 2023 more than 200,000 COVID-19 cases, while it reported 2246 cases of influenza during the 2023–2024 cold season [14]; however, the difference could be influenced by an easier reporting system for the novel coronavirus infection. COVID-19 continued to produce waves year-long, with very few cases during summer, while influenza maintained the expected pattern of seasonality.

Patients hospitalized for COVID-19 were significantly older than influenza ones, and especially patients with severe disease. Atamna et al. and Donnino et al. published similar data [15,16], but the age gap in our database is largest. Also, when comparing to data from previous COVID-19 waves, our patients are older than patients from initial waves [17]. Related to influenza, the median age of hospitalized patients was similar to that reported by Atamna et al., and similar to previous Romanian cohorts published by our team [3], but younger than patients in other reports [18]. As expected, the older age of COVID-19 patients was reflected in ACCI, which was higher than for Influenza A patients, similar to the research of Atamna et al [15]. However, the analysis of the standard Charlson comorbidity index revealed that the two cohorts shared the same median value.

Female sex was more prevalent in hospitalized patients for both infections, but male sex was more prevalent for severe and critical Omicron patients. This finding is consistent across literature and across COVID-19 waves, stating that due to immunosenescence and presumed hormonal factors, male patients are at increased risk of unfavorable outcomes in SARS-CoV-2 infection [17,19].

Information regarding vaccination was scarce in our database, but most Influenza A hospitalizations in this study were in unvaccinated patients. Romania constantly reported low vaccination uptake in the general population [20,21], reflected in a significant number of hospitalizations. Reported COVID-19 vaccination in the database is higher than for influenza, though a high number of these patients were vaccinated only in 2021 for previous variants of SARS-CoV2 and antibodies are expected to have waned [22]. Safety of simultaneous vaccination for both infections was proven to be safe [23], but in this database, causality between vaccinations and hospitalizations cannot be addressed.

Related to risk factors, we analyzed the conditions comprised in the ACCI. We found that age-related comorbidities such as cancer, dementia, and hemiplegia were more frequent in the older cohort of COVID-19 patients. Cardiovascular comorbidities and diabetes mellitus were frequent in both groups, all representing risk factors for hospitalization and severe disease in many infections. Obesity, a potent risk factor for hospitalization and severe outcome during the first waves of SARS-CoV-2 pandemic [24,25], was more frequent in severe Influenza A patients, as well as COPD and asthma. In a paper by Dupont et al., asthma was also more prevalent in influenza patients than in COVID-19 [26]. We could not find definitive data on COPD’s compared incidence between the two infections, but since most COPD exacerbations are caused by viral infections [27], and a significant percentage of both groups had either asthma or COPD, increased awareness for these pulmonary comorbidities is necessary. A positive aspect of our study is that pregnant and puerperal women, as well as solid-organ transplant recipients, were rare and none experienced severe outcomes. This may be attributed to a heightened vigilance and specialized management these high-risk groups receive both as out-patients and in-hospital settings.

Although previous reports showed that the time from symptom onset until hospital presentation is longer for COVID-19 [28], in our database, the duration until hospital presentation was short and identical for both infections.

Incidence of complications, such as respiratory failure, need for respiratory support, or ICU admittance, is not uniform in the literature due to several factors, such as the variant of SARS-CoV-2, study population selection, availability of specialized respiratory support such as CPAP on wards, and local protocols of hospitals. Atamna et al. [15] reported that 61% of COVID-19 and 40% of influenza patients needed supplemental oxygen, 7% of Omicron and 3% of flu subjects needed invasive mechanical ventilation, and 14% of Omicron and only 2% of influenza patients were admitted to the intensive care unit in a study carried out on an adult population between December 2021 and January 2022. In a study comparing SARS-CoV-2-infected patients from the beginning of the pandemic with influenza patients from 2017/2018 and 2018/2019 influenza seasons, Pawelka et al. found acute respiratory failure in 66.9% of COVID-19 patients and 36.84% of Influenza A patients, with ICU admission in 8.45% of COVID-19 patients and 9.02% of Influenza A patients [18]. Piroth et al. [29], in a large nationwide retrospective study comparing COVID-19 patients from March–April 2020 with influenza patients from the 2018/2019 season, including both adults and children, identified acute respiratory failure in 27.2% of COVID-19 patients and 17.4% of influenza patients, mechanical ventilation in 9.7% of COVID-19 patients and 4% of influenza subjects, and ICU admission in 16.3% of COVID-19 patients and 10.8% of influenza patients. We are not aware of more recent comparative analyses between the two infections. In our database, the younger influenza population exhibited a more frequent need for supplemental oxygen or CPAP than the elderly Omicron patients; the latter, on the other hand, were more likely to be admitted to the ICU or need mechanical ventilation. Notably, patients without respiratory failure exhibited a low risk of other complications. Pulmonary embolism was more frequent in Omicron patients in our study. Pulmonary embolism has a higher prevalence in COVID-19 patients compared to the general population and those with influenza [30]. In our hospital, all COVID-19 patients undergo routine D-dimer testing as part of active screening for thrombotic complications. Other complications, such as onset of atrial fibrillation, stroke, and digestive hemorrhage, although rare in our cohort, happened more often in COVID-19 patients. A high and equivalent number of patients from both groups were prescribed antibiotic therapy, a much more frequent event than in other analyzed papers [15,18].

Interesting results emerged from the multivariate analysis, showing that the odds of ICU admission, CPAP use, or intubation were higher for influenza compared to Omicron SARS-CoV-2, although statistical significance was reached only for CPAP. Laboratory parameters that significantly influenced outcomes in both viral infections included elevated C-reactive protein, LDH, and NLR, as well as a high ACCI score. CRP and LDH were associated with increased ICU admission risk in a study conducted early in the COVID-19 pandemic [31]. A study comparing Omicron and Influenza also found greater CPAP use in influenza patients than in COVID-19 patients. Similar to our findings, ICU admission rates were comparable between the two groups, with no statistically significant difference [32].

Since the beginning of the COVID-19 pandemic, lymphopenia was described more often and more specific in association with the novel coronavirus than influenza [33]. More recent papers also reported a lower number of leucocytes and a higher value of the CRP for SARS-CoV-2 infection [15,18], but influenza patients in our database expressed lower number of lymphocytes and higher CRP. Also, the CBC-derived biomarkers, especially NLR, were also elevated in both infections. Interestingly, NLR on admission was very high in patients who died, supporting a prognostic role for this cheap inflammation marker, which was also reported by other researchers [34]. Whether evolving COVID-19 variants have reduced capacity to induce lymphopenia is a matter of future research.

Our study indicates the presence of radiological differences between COVID-19 and Influenza A. We found that radiological lesions, especially ground-glass opacities, were significantly more frequent in COVID-19 cases compared to influenza, as was previously described [35]. Omicron has also shown a higher prevalence of bilateral consolidations and viral pneumonitis while, conversely, unilateral consolidations were more common in Influenza A group. Similar findings were recently reported [36].

The in-hospital mortality rates observed in our study are consistent with recent publications, although the difference between COVID-19 and Influenza A did not reach statistical significance in our analysis. For instance, in two papers published by Xie Y et al., in which they compared risk of death from COVID-19 vs. seasonal influenza in two consecutive seasons. They found a mortality rate of 5.98% for COVID-19 and 3.16% for influenza during the 2022/2023 cold season and 5.7% for COVID-19 and 3.04% for influenza during the 2023/2024 cold season [37,38]. Multivariate regression on our data also revealed COVID-19 had a higher OR for in-hospital death, even if more patients with influenza had severe disease and the OR for ventilation was higher in influenza, which could be explained by the population with COVID-19 being older and frailer.

We acknowledge that a large number of univariate comparisons were conducted, which increases the risk of type I statistical error. Given the exploratory nature of the study, we chose not to apply corrections such as Bonferroni or FDR. However, *p*-values—especially those close to 0.05—should be interpreted cautiously and primarily as signals warranting further investigation.

Our findings suggest that individuals with elevated inflammatory markers (CRP, LDH, and NLR), impaired renal function, and higher comorbidity burden (ACCI ≥ 4) are at significantly increased risk of ICU admission, respiratory support, or in-hospital death. These risk factors, consistent across both influenza and Omicron COVID-19, can help guide clinical decision making. We recommend that patients with these risk profiles be prioritized for seasonal vaccination campaigns, considered for early antiviral therapy when indicated, and receive closer ambulatory monitoring during respiratory virus season. Integrating such risk stratification into public health planning may enhance resource allocation and reduce hospitalization burden in future viral seasons.

There are several limitations to the present study. First, in our retrospective database, information regarding vaccination was scarce, and therefore, the vaccination status, which might have had an influence on disease outcomes, could not be considered in the statistical analysis. Second, not all patients were diagnosed with molecular tests, but guideline adherence for diagnosis was ensured. Third, mortality data was collected as in-hospital death, because the follow-up after discharge was not performed. Fourth, while we recorded CBC data, inflammation markers, and imaging findings, microbiologic data was available for very few patients. Additionally, the majority of cases received empirical antibiotic therapy. Consequently, in this retrospective analysis, we could not reliably stratify bacterial superinfection. Finally, this was a single-center study conducted in an infectious diseases hospital. While this ensured consistency in diagnostic and therapeutic approaches, it may limit the external generalizability of our findings to other settings with different patient populations, healthcare systems, or protocols. A strength of being a single-center study resides in using a significant amount of uniform data, from a single hospital; our findings provide practical insights into clinical and laboratory presentation of patients with unfavorable outcomes; and our study is one of the few to study the two infections in a concomitant recent timeframe with relevance to clinical decision making.

In conclusion, the data we presented indicates that seasonality, demographics, prevalence of comorbidities, and clinical presentation are different between the COVID-19 Omicron variant and Influenza A. Patients with COVID-19 admitted to the hospital tended to be older, with higher ACCI, more frequent complications, and a higher in-hospital mortality than those with Influenza A. At the same time, Influenza A risk factors remained largely unchanged in the last seasons, but in younger patients with comorbidities, the risk of needing advanced respiratory support or intensive care unit admission is higher. Common laboratory parameters continue to be useful for predicting poor outcomes.

## Figures and Tables

**Figure 1 microorganisms-13-01836-f001:**
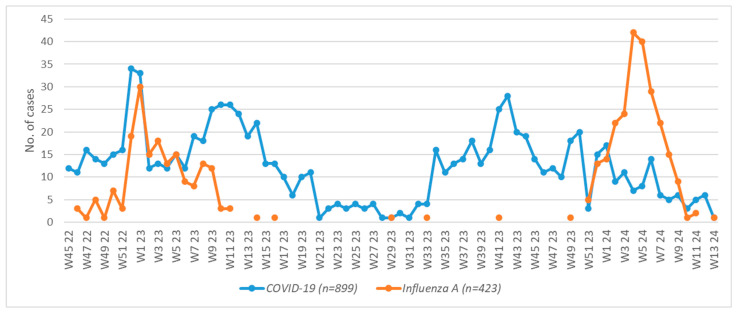
Weekly hospital admissions for Influenza A and COVID-19, stratified by year and week number. Influenza outbreaks show a distinct seasonal pattern with sharp peaks, while COVID-19 admissions occurred year-round with broader wave patterns. Epidemic curves did not overlap.

**Table 1 microorganisms-13-01836-t001:** Baseline characteristics and comorbidities of study groups.

Variables	COVID-19 (*n* = 899)	Type A Influenza (*n* = 423)	*p*-Value
Age (years), median (IQR)	74 (65–82)	65 (45–78)	<0.001
Sex (F), *n* (%)	501 (55.73)	255 (60.28)	0.118
ACCI, median (IQR)	5 (3–6)	3 (1–5)	<0.001
CCI without age, median (IQR)	1 (0–3)	1 (0–2)	<0.001
** *Comorbidities* **			
Active cancer, *n* (%)	103 (11.46)	19 (4.49)	<0.001
Asthma, *n* (%)	60 (6.67)	41 (9.69)	0.054
Atrial fibrillation, *n* (%)	176 (19.58)	68 (16.08)	0.126
AIDS, *n* (%)	9 (1)	2 (0.47)	0.518
Connective tissue disease, *n* (%)	13 (1.45)	7 (1.65)	0.772
Chronic kidney disease, *n* (%)	97 (10.79)	41 (9.69)	0.543
Chronic hepatitis, *n* (%)	22 (2.45)	13 (3.07)	0.508
Congestive heart failure, *n* (%)	213 (23.69)	84 (19.86)	0.119
COPD, *n* (%)	116 (12.9)	68 (16.08)	0.12
Dementia, *n* (%)	106 (11.79)	34 (8.04)	0.039
Diabetes mellitus, *n* (%)	238 (26.47)	103 (24.35)	0.41
Hemiplegia, *n* (%)	80 (8.9)	18 (4.26)	0.003
Hypertension, *n* (%)	626 (69.63)	242 (57.21)	<0.001
History of myocardial infarction, *n* (%)	59 (6.56)	21 (4.96)	0.256
History of stroke or TIA, *n* (%)	145 (16.13)	51 (12.06)	0.052
Ischemic heart disease, *n* (%)	238 (26.47)	97 (22.93)	0.167
Leukemia, *n* (%)	16 (1.78)	2 (0.47)	0.056
Lymphoma, *n* (%)	10 (1.11)	2 (0.47)	0.358
Liver cirrhosis, *n* (%)	11 (1.22)	4 (0.95)	0.786
Obesity, *n* (%)	195 (21.69)	105 (24.82)	0.205
Peptic ulcer disease, *n* (%)	45 (5.01)	16 (3.78)	0.323
Peripheral vascular disease, *n* (%)	92 (10.23)	30 (7.09)	0.066
Pregnant/post-natal, *n* (%)	21 (2.34)	13 (3.07)	0.43
SOT recipient, *n* (%)	4 (0.44)	5 (1.18)	0.155
Seasonal influenza vaccination, *known status n = 141* (%)		Y: 24 (17.02) N: 117 (82.98)	
COVID-19 vaccination, *known status n = 425* (%)	Y: 327 (76.95) N: 98 (23.95)		

IQR—interquartile range; ACCI—age-adjusted Charlson comorbidity index; CCI—Charlson comorbidity index; AIDS—acquired immune deficiency syndrome; SOT—solid organ transplantation; COPD—chronic obstructive pulmonary disease; TIA—transient ischemic attack; Y—yes; N—no.

**Table 2 microorganisms-13-01836-t002:** Laboratory parameters on admission and radiologic appearance for COVID-19 and Influenza A.

	COVID-19 (*n* = 899)	Influenza A (*n* = 423)	*p*-Value
** *Laboratory findings, median (IQR)* **			
Leucocyte ^1^	6.5 (4.88–8.9)	6.2 (4.5–8.26)	0.028 [n1 = 895, n2 = 421]
Neutrophils ^1^	4.65 (3.14–6.94)	4.65 (2.98–6.6)	0.322 [n1 = 895, n2 = 421]
Lymphocytes ^1^	0.98 (0.65–1.49)	0.91 (0.59–1.38)	0.049 [n1 = 895, n2 = 421]
Monocytes ^1^	0.56 (0.36–0.78)	0.49 (0.34–0.7)	0.008 [n1 = 895, n2 = 421]
Thrombocytes ^1^	192 (147–247)	183 (147–233)	0.071 [n1 = 895, n2 = 421]
NLR	4.55 (2.58–8.83)	5.22 (2.75–9.74)	0.215 [n1 = 894, n2 = 419]
dNLR	2.74 (1.63–4.89)	3.17 (1.78–5.18)	0.12 [n1 = 894, n2 = 419]
PLR	188.52 (127.42–293.24)	198.32 (132.8–308.04)	0.275 [n1 = 894, n2 = 419]
SII	863.64 (446.66–1805.02)	942.53 (459.36–1666.23)	0.807 [n1 = 894, n2 = 419]
SIRI	2.4 (1.21–4.92)	2.55 (1.19–4.92)	0.875 [n1 = 894, n2 = 419]
MLR	0.53 (0.34–0.83)	0.55 (0.35–0.84)	0.761 [n1 = 894, n2 = 419]
C-reactive protein	3.99 (1.47–9.23)	4.52 (1.85–9.62)	0.076 [n1 = 897, n2 = 419]
Procalcitonin	0.5 (0.33–1.32)	0.45 (0.08–2.58)	0.04 [n1 = 208, n2 = 94]
Fibrinogen	415.33 (330.21–497.94)	398.95 (322.84–474.36)	0.353 [n1 = 595, n2 = 165]
LDH	210 (174.5–265.5)	232 (184.25–283.75)	0.023 [n1 = 623, n2 = 102]
Creatinine	0.9 (0.73–1.17)	0.86 (0.69–1.12)	0.021 [n1 = 897, n2 = 420]
** *Radiological appearance, n (%)* **			
Ground glass opacities	235 (26.14)	48 (11.35)	<0.001
Consolidation, unilateral	116 (12.9)	72 (17.02)	0.046
Consolidation, bilateral	163 (18.13)	56 (13.24)	0.026
Interstitial pattern	491 (54.62)	197 (46.57)	0.006
No radiological lesions	216 (24.02)	128 (30.26)	0.016

^1^—×103; IQR—interquartile range; NLR—Neutrophils-to-lymphocytes ratio; dNLR—derived neutrophils-to-lymphocytes ratio, neutrophils/[leucocytes-neutrophils]; PLR—platelets-to-lymphocytes ratio; SII—systemic inflammation index, neutrophils×platelets/lymphocytes; SIRI—systemic inflammation response index, neutrophils×monocytes/lymphocytes; MLR—monocytes-to-lymphocytes ratio; n1—number of COVID-19 patients; n2—number of influenza A patients; Patients exhibited one or more of the radiologic lesions described.

**Table 3 microorganisms-13-01836-t003:** Clinical forms of disease for influenza type A and COVID-19: sex and age distribution.

Disease	Clinical Form	Number (%)	Age, Median	Sex, F
Influenza A	Severe	297/423 (70.21)	71	164 (55.21)
	Nonsevere	126/423 (29.79)	37	91 (72.2)
COVID-19	Mild	185/899 (20.58)	57	126 (68.1)
	Medium	335/899 (37.26)	74	201 (60)
	Severe	261/899 (29.03)	78	116 (44.44)
	Critical	118/899 (13.13)	80	58 (49.15)

F—female sex.

**Table 4 microorganisms-13-01836-t004:** Complications and outcomes in hospitalized cohorts of COVID-19 vs. Influenza A.

Variables	COVID-19 (*n* = 899)	Influenza A (*n* = 423)	*p*-Value
Duration of hospitalization (days), *median (IQR)*	6 (4–9)	6 (4–8)	0.082
Days before hospital admission, *median (IQR)*	3 (2–6)	3 (2–5)	0.282
Antibiotic treatment, *n* (%)	713 (79.31)	347 (82.03)	0.278
**Acute respiratory failure, *n* (%)**	501 (55.73)	266 (62.88)	0.014
Noninvasive ventilatory support, *n* (%)	150 (16.69)	77 (18.2)	0.495
Invasive ventilatory support, *n* (%)	57 (6.34)	20 (4.73)	0.243
Respiratory failure and other complications, *n* (%)	155 (30.94)	89 (33.46)	0.244
No respiratory failure, but other complications, *n* (%)	63 (15.83)	22 (14.01)	0.065
**ICU admission, *n* (%)**	106 (11.79)	40 (9.46)	0.206
Days of ICU stay, *median (IQR)*	7 (3–16.75)	7 (4–9)	0.867
Newly diagnosed atrial fibrillation, *n* (%)	38 (4.23)	15 (3.55)	0.556
Acute renal failure, *n* (%)	150 (16.69)	93 (21.99)	0.02
Pulmonary embolism, *n* (%)	27 (3)	8 (1.89)	0.24
Digestive hemorrhage, *n* (%)	14 (1.56)	3 (0.71)	0.202
Stroke, *n* (%)	14 (1.56)	3 (0.71)	0.202
Acute myocardial infarction, *n* (%)	4 (0.44)	1 (0.24)	1
Acute myocarditis, *n* (%)	1 (0.11)	0 (0)	1
Acute myositis, *n* (%)	1 (0.11)	0 (0)	1
Encephalitis, *n* (%)	1 (0.11)	0 (0)	1
**Deceased, *n* (%)**	51 (5.67)	14 (3.3)	0.064

IQR—interquartile range; ICU—intensive care unit.

**Table 5 microorganisms-13-01836-t005:** Characteristics of deceased COVID-19 and Influenza A patients.

Variables	COVID-19 (*n* = 51)	Type A Influenza (*n* = 14)	*p*-Value
Age (years), *median (IQR)*	80 (72–89)	75 (65.5–83.25)	0.18
Sex (F), nr (%)	27 (53)	7 (50)	0.845
ACCI, *median (IQR)*	7 (5–8)	7 (6–8)	0.434
Length of stay (days), *median (IQR)*	13 (7–22)	10 (6.5–29.25)	0.823
Days of ICU stay, *median (IQR)*	11 (3–21)	8 (5–23)	0.778

IQR—interquartile range; ACCI—age-adjusted Charlson comorbidity index.

**Table 6 microorganisms-13-01836-t006:** Multivariate regression model with splines for ICU admission, CPAP, intubation and death OR, adjusted for sex, ACCI, CRP, creatinine, LDH, and NLR.

Variable	ICU Admission OR (95% CI), *p*	CPAP OR (95% CI), *p*	Intubation OR (95% CI), *p*	Death OR (95% CI), *p*
**COVID-19 vs. Influenza A**	0.69 (0.36–1.35) *p* = 0.283	**0.56 (0.32–0.99)** ***p* = 0.044**	0.65 (0.26–1.63) *p* = 0.362	1.35 (0.69–2.63) *p* = 0.381
Sex (Male vs. Female)	1.02 (0.65–1.6) *p* = 0.939	1.01 (0.67–1.5) *p* = 0.970	0.72 (0.4–1.26) *p* = 0.251	0.87 (0.51–1.46) *p* = 0.593
ACCI ≥ 4	1.62 (0.9–3) *p* = *0.117*	1.25 (0.76–2.07) *p* = 0.382	**2.85 (1.21–7.91)** ***p* = 0.026**	**5.81 (2.31–19.55)** ***p* < 0.001**
CRP ≥ 4.225	**2.11 (1.23–3.73)** ***p* = 0.008**	**3.01 (1.89–4.92)** ***p* < 0.001**	3.10 (1.45–7.44) *p* = 0.006	**5.86 (2.9–13.52)** ***p* < 0.001**
Creatinine ≥ 0.89	1.23 (0.76–2.01) *p* = 0.398	1.21 (0.8–1.86) *p* = 0.372	1.34 (0.73–2.55) *p* = 0.357	
LDH ≥ 213	**5.77 (3.34–10.54)** ***p* < 0.001**	**3.61 (2.35–5.64)** ***p* < 0.001**	**5.83 (2.74–14.42)** ***p* < 0.001**	
NLR (Neu/Lym) ≥ 4.67	**4.29 (2.45–7.93),** ***p* < 0.001**	**2.86 (1.83–4.55)** ***p* < 0.001**	**3.56 (1.69–8.44)** ***p* = 0.002**	

ACCI—age-adjusted Charlson comorbidity index; CRP—C-reactive protein; LDH—Lactate Dehydrogenase; NLR—neutrophil-to-lymphocyte ratio. Multivariate regression model using splines for continuous variables. Model with dichotomized predictors available in Appendix A. Statistically significant odds ratios are bolded.

## Data Availability

Data used was retrieved from the database of the Teaching Hospital of Infectious Disease, Cluj-Napoca, Romania. The dataset is not public, but may be made available on request; please contact the corresponding authors.

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
