# Peer review of "Influenza A vs. COVID-19: A Retrospective Comparison of Hospitalized Patients in a Post-Pandemic Setting"

_microorganisms, 2025, doi:10.3390/microorganisms13081836_

Round 1
Reviewer 1 Report
Comments and Suggestions for Authors
1. Single data source, limited extrapolation (page 2, lines 60-61)
The data in the paper all come from a hospital in Romania, and do not include data from other medical institutions, so the conclusions are difficult to generalize to other regions or hospitals.
2. A large amount of vaccine data is missing, but causal inferences are made (page 5, Table 1; page 11 discussion section)
The missing rate of vaccination information exceeds 50%, but the author still clearly infers that vaccines have an important impact on hospitalization rates. This is inappropriate and should be deleted or modified.
3. The confounding effect of the trend of hospitalization time is not considered (pages 2-3, lines 62-64; page 9, Table 6)
The study spanned 17 months, but the analysis did not control the hospitalization period (season or epidemic wave), which may cause confounding bias and affect the reliability of the research conclusions. Time variables should be added or clearly stated.
4. A large number of hypothesis tests were not corrected, and the risk of false positives was high (Table 2 on page 6; Table 4 on page 8)
More than 30 variables were tested separately, but no Bonferroni or FDR correction was performed, which increased the risk of statistical error and should be corrected or explained.
5. The interpretation of mortality rate is misleading (Table 4 on page 8; Discussion section on page 13)
The mortality rate of COVID-19 is 5.67%, and that of influenza is 3.31% (p=0.064). The author's direct interpretation as "no difference" is not rigorous.
6. The two sets of model analysis are redundant, and the necessity is not clearly explained (page 4, lines 110-114; Table 6 on page 9)
Two sets of models (spline and median binary classification) were constructed for each result, and the purpose or difference was not clearly explained. It is recommended to simplify the model and keep only the main model.
7. Subjective inference of vaccine effectiveness (Discussion on page 11)
Subjective expressions such as "vaccination should significantly affect hospitalization" appear in many places in the article, which lack empirical data support and should be avoided.
8. Use of subjective evaluative language (Discussion on pages 12-13)
Expressions such as "interestingly", "highlights", and "predictive" are recommended to be modified to more objective and accurate expressions.
9. Tables and diagrams contain too much information and do not highlight the key points (Table 6 on page 9, Table 4 on pages 7-8, Figure 1)
Some tables are too dense and the charts do not highlight the time trend.
Reviewer 2 Report
Comments and Suggestions for Authors
Dear Authors,
This study examined the critical issue of comparing the two predominant viral respiratory infections, type A influenza and COVID-19 (Omicron variant), with respect to their clinical progression, risk factors for severe illness, complications, and in-hospital mortality. The authors utilized data from a single clinical center in Romania from November 2022 to March 2024.
The study was described as a "retrospective cohort study," which is both imprecise and methodologically incorrect. A cohort study typically involves observing a population of healthy individuals or individuals exposed to a specific risk factor over time to evaluate the impact of this exposure on disease development. However, in this study, the analysis was conducted on data from patients already hospitalized with a diagnosed infection; thus, it does not constitute a cohort in an epidemiological sense. Furthermore, this was not a case-control study, which would necessitate the presence of a control group (e.g., healthy individuals or those with another disease). This was a retrospective comparative analysis of two groups of patients hospitalized due to different viral infections. This inconsistency should be rectified in both the Abstract and the "Materials and Methods" section to prevent misinterpretation of the nature of the study by readers.
The discussion presented in the publication is comprehensive and well founded within the context of current medical knowledge. The authors accurately interpreted the results of the statistical analyses, emphasizing the differences in the clinical course and prognosis between patients with COVID-19 (Omicron variant) and those with type A influenza. However, it is advisable to augment the discussion with practical clinical recommendations that could emerge from the multivariate analyses. Specifically, because the authors identified particular risk factors and patient groups it would be beneficial to consider whether specific preventive measures should be implemented for these groups, such as intensifying seasonal vaccination, earlier qualification for antiviral treatment, or prioritizing them for ambulatory care during the infection season. Such recommendations would enhance the practical value of this study and could significantly contribute to health policy planning in the context of future infection seasons.
Good luck
Round 2
Reviewer 1 Report
Comments and Suggestions for Authors
There is no suggestion for this reversion.